# Examining the Association between Coffee Intake and the Risk of Developing Irritable Bowel Syndrome: A Systematic Review and Meta-Analysis

**DOI:** 10.3390/nu15224745

**Published:** 2023-11-10

**Authors:** Jasmine Yiling Lee, Chun Yi Yau, Caitlin Yuen Ling Loh, Wei Shyann Lim, Seth En Teoh, Chun En Yau, Clarence Ong, Julian Thumboo, Vikneswaran S. O. Namasivayam, Qin Xiang Ng

**Affiliations:** 1NUS Yong Loo Lin School of Medicine, National University of Singapore, Singapore 117597, Singapore; 2Saw Swee Hock School of Public Health, National University of Singapore, Singapore 117549, Singapore; 3Health Services Research Unit, Singapore General Hospital, Singapore 169608, Singapore; 4SingHealth Duke-NUS Medicine Academic Clinical Programme, Duke-NUS Medical School, Singapore 169857, Singapore; 5Department of Gastroenterology and Hepatology, Singapore General Hospital, Singapore 169856, Singapore

**Keywords:** irritable bowel syndrome (IBS), coffee, caffeine, risk

## Abstract

Irritable bowel syndrome (IBS) is a highly prevalent disorder of brain–gut interaction with a significant impact on quality of life. Coffee is a widely consumed beverage with numerous bioactive compounds that have potential effects on human health and disease states. Current studies on the effect of regular coffee consumption on the risk of developing IBS symptoms have yielded conflicting results. This systematic review and meta-analysis aimed to determine whether coffee intake is associated with developing IBS. A systematic literature search was performed in three electronic databases, namely PubMed, EMBASE, and The Cochrane Library, from inception until 31 March 2023. All original studies reporting associations between coffee intake and IBS were considered for inclusion. Odds ratios (ORs) were calculated for each study, and estimates were pooled, and where appropriate, 95% confidence intervals (95% CI) and *p*-values were calculated. Eight studies comprising 432,022 patients were included in the final meta-analysis. Using a fixed-effects model, coffee drinkers (any intake) had a reduced likelihood of developing IBS compared to controls, with a pooled OR of 0.84 (95% CI: 0.80 to 0.84). Sensitivity analysis confirmed the stability of the estimates. Future research should prioritise prospective cohort studies that are robust and closely track the development of incident IBS in previously healthy individuals.

## 1. Introduction

Irritable bowel syndrome (IBS) is a highly prevalent disorder of brain–gut interaction with a significant impact on quality of life and social functioning and a considerable socioeconomic cost. Reports suggest that between 5% and 10% of the global populace grapple with IBS at any given time [1]. The detrimental impact on quality of life due to IBS is comparable to that caused by organic illnesses, including inflammatory bowel disease [2]. Furthermore, IBS is often associated with diminished work productivity, reduced participation in social activities, and increased absenteeism [3,4].

One key factor that has been consistently associated with IBS is diet. Diet has been implicated in the pathophysiology of IBS as well as disease flares. A significant proportion of IBS patients experience food-related symptoms that are associated with consuming or eliminating certain foods. Food-related symptoms are associated with high symptom burden, reduced quality of life, increased healthcare utilisation and reduced nutrient intake [5]. While randomised clinical trials have demonstrated the potential benefits of dietary modifications in alleviating IBS symptoms for some [6,7], and dietary interventions, such as the low fermentable oligo-, di-, monosaccharides and polyols (FODMAP) diet have been recommended as first-line treatments for IBS [7], observational studies have further hinted at the role of the diet in the onset of IBS in individuals who were previously healthy [8]. A previous study from Greece found that greater adherence to a Mediterranean-style diet, typically rich in whole grains, fruits, vegetables, nuts, and seeds, is inversely associated with the occurrence of IBS among children and adolescents [8]. The gut microbiome, a pivotal player in human health, is modulated by dietary habits and food intake [9]. However, the exact dietary patterns and broader health implications of these microbiome alterations remain to be thoroughly explored.

When examining human lifestyle, environmental and nutritional factors, coffee quickly emerges as one of the most widely consumed beverages in the world [10]. Coffee is one of the most traded global commodities and a cultural phenomenon that has profoundly influenced social interaction and human history. Coffee consumption has been linked to a range of health benefits, including protective effects against neurological and metabolic diseases and certain cancers [11,12]. A previous umbrella review on coffee intake and various health outcomes found that drinking coffee is typically safe at expected human consumption levels, and consuming three to four cups daily appears to offer the most health benefits, reducing the risk of specific cancers as well as neurological, metabolic, and liver diseases [11]. While lab-based and epidemiologic studies have revealed that coffee can have several impacts on the digestive system [13], including antioxidant, anti-inflammatory, and antiproliferative actions, its causal effects on gut health outcomes are still incompletely understood.

Multiple studies have implicated coffee intake in the occurrence of IBS, but the results have not been consistent [14,15,16]. Theoretically, coffee contains caffeine and other bioactive compounds with pleiotropic effects on multiple pathways, ranging from the gut microbiome [16,17] and bile acid metabolism [18,19] to intestinal permeability, gastrointestinal transit [20], and even the central nervous system [13,21], all of which have potential implications in IBS pathogenesis. Nevertheless, the relationship between coffee consumption and the risk of developing IBS remains controversial.

Given the ubiquity of coffee, clarifying the role of coffee and its constituents on the risk of incident IBS may identify potential primary preventive strategies centred on modifying coffee intake. As there has not been a systematic review or meta-analysis until now, this study aims to determine whether there is an association between coffee intake and the likelihood of developing IBS.

## 2. Methods

This systematic review and meta-analysis adhered to the Preferred Reporting Items for Systematic Reviews and Meta-Analyses (PRISMA) guidelines [22]. The study protocol was prospectively registered in the International Prospective Register of Systematic Reviews (PROSPERO), registration number CRD42023427985.

A systematic literature search was performed in three electronic databases, namely PubMed, EMBASE, and The Cochrane Library, from the databases’ inception up until 31 March 2023. The search strategy was developed in consultation with a medical information specialist (Medical Library, National University of Singapore), and key search terms such as “irritable bowel syndrome” and “coffee” were used in the search strategy. No restrictions on date, language, or subject were implemented in the database search. The detailed search strategy can be found in the Appendix A. Content experts were also consulted for additional references, and references of sources were hand-searched to identify additional relevant articles. Articles were viewed through Rayyan (Qatar Computing Research Institute, Doha, Qatar), and duplicates were identified and removed.

The selection of articles for inclusion was conducted by four researchers (J.Y.L., C.Y.Y., S.E.T, and C.Y.L.L.). Each article was reviewed by at least two researchers blinded to each other’s decision. Disputes were resolved through consensus from the senior author (Q.X.N.). The predefined criteria for inclusion were as follows: (1) articles reporting associations between coffee intake and IBS, (2) original articles (randomised controlled trials (RCTs), case–control studies, cohort and cross-sectional studies), and (3) articles written or translated into the English language. The criteria for exclusion were: (1) studies that analysed caffeine, tea, or other dietary intake without any analysis of coffee alone, (2) studies that only reported the exacerbation of symptoms in pre-existing IBS patients, and (3) commentaries, consensus-based guidelines, case reports, case series, review articles, and conference abstracts.

The data were carefully extracted and catalogued into a Microsoft Excel (Microsoft Corp, Albuquerque, NM, USA) spreadsheet by four authors (J.Y.L., C.Y.Y., S.E.T, and C.Y.L.L.) to ensure precision. They were subjected to double coding, affirming the accuracy of the captured information. The abstracted data encompassed various study aspects such as details about the authors, publication year, geographical context, and characteristics of the study population, including sample size, demographic details, and diagnostic criteria. Additionally, primary outcomes, like the diagnosis of IBS among individuals who consume coffee, were recorded. Continuous variables were documented with mean and standard deviation (SD) values. When such data were not directly available, they were derived from the median and range (or interquartile range) through appropriate mathematical transformations [23]. For categorical variables, both frequency and percentages were noted.

To assess the potential risk of bias in the included studies, two reviewers (J.Y.L. and C.Y.Y.) independently employed the Newcastle–Ottawa Scale (NOS) for cross-sectional and cohort studies [24]. This tool thoroughly evaluated each study across three distinct domains: the selection process for the research population, comparability between study groups, and assurance of accurate results. Studies scoring seven or above were deemed high-quality. In instances where discrepancies arose, they were addressed and resolved through discussions with the senior author (Q.X.N.).

All data analyses were conducted using R 4.0.3 (R Foundation for Statistical Computing, Vienna, Austria) [25]. Each study was closely reviewed, and the primary outcome measure of interest was the likelihood of developing IBS compared to a control group. Odds ratios (ORs) were calculated for each study, and estimates were pooled, and where appropriate, 95% confidence intervals (95% CI) and *p*-values were calculated. Two-tailed statistical significance was set at *p*-value < 0.05. The Cochran Q test and *I*^2^ statistics were utilised to quantify heterogeneity amongst the different studies pooled. *I*^2^ value thresholds of 25%, 50% and 75% signified low, moderate and high heterogeneity, respectively [26]. Publication bias was assessed via visual inspection of funnel plots and substantiated using Egger’s regression test [27].

## 3. Results

### 3.1. Literature Retrieval

A total of 187 publications were initially identified by searching the PubMed, Embase and Cochrane databases after removing duplicates. As shown in Figure 1, after reviewing the titles and abstracts, 55 articles were selected for full-text reading, of which the full-text for 16 articles could not be located despite attempts to contact the authors. After considering our inclusion and exclusion criteria, eight studies [14,15,16,28,29,30,31,32], with a collective total of 432,022 participants, were included in the final meta-analysis. The risk of bias in the included studies was moderate to high. This was particularly a concern among the cross-sectional studies reviewed [14,15,16,31] due to the lack of a representative sample and absent confounder analysis. The individual ratings and breakdowns can be found in the Appendix A.

The key study characteristics are summarised in Table 1. Overall, the regions where the studies were performed were as follows: Asia (*n* = 6), the United Kingdom (*n* = 1) and Africa (*n* = 1). Six studies [15,16,28,29,30,31] used the established ROME III criteria to diagnose IBS, while one [30] identified cases through International Classification of Disease—10th revision (ICD-10) codes, and another [14] used the earlier ROME II criteria for IBS diagnosis. There were no data on the specific occurrence of each IBS subtype among the primary studies.

In terms of exposure, coffee intake was defined rather inconsistently across the primary studies, with some adopting a binary approach (coffee consumer or abstainer) as opposed to three-level or finer categorisations, as seen in Table 1.

### 3.2. Coffee Intake and Risk of Developing IBS

Given the small number of studies available and the scenario of a high-quality study with a large sample size (Wu et al., 2023 [32]) compared to other low-quality studies with smaller sample sizes (Khademolhosseini et al., 2011 [14], Basandra et al., 2014 [15], Guo et al., 2015 [16] and El Sharawy et al., 2022 [31]), a fixed-effects model was chosen for greater precision of estimates (as it gives more weight to larger and more precise studies) [33]. We also assumed a common true effect for coffee across the studies.

As shown in Figure 2, compared with controls, the exposed group (any coffee intake) appeared to have a reduced likelihood of developing IBS, with a pooled OR of 0.84 (95% CI: 0.80 to 0.88). Heterogeneity was high (*I*^2^ > 75%), likely because of the fact the pooled studies had different approaches to classifying coffee intake and different methods were used to define the exposure (coffee intake) and outcome (IBS). It is difficult to accurately estimate the between-study variance due to the small number of studies.

Since Koochakpoor and colleagues [30] had ORs reported for two different coffee intake levels (monthly as opposed to weekly), a sensitivity analysis was conducted with the monthly coffee intake level. As seen in Figure 3, the pooled OR remained comparable (pooled OR: 0.84, 95% CI: 0.80 to 0.88), affirming the stability of the estimates.

When examining the likelihood of developing IBS for coffee drinkers versus non-coffee drinkers or abstainers, Wu et al.’s [32] study (a longitudinal cohort study with a large sample size) was included in this analysis (shown in Figure 4); coffee drinkers had an overall lower likelihood of developing IBS still, with a pooled OR of 0.83 (95% CI: 0.79 to 0.87).

When Wu et al.’s [32] study was excluded from this analysis (shown in Figure 5), the effect was no longer observed, with a pooled OR of 1.00 (95% CI: 0.74 to 1.36) as compared to the pooled OR 0.83 (95% CI: 0.79 to 0.87).

## 4. Discussion

This systematic review and meta-analysis of eight studies, including 432,022 individuals, has summarised the published evidence on coffee intake and the risk of incident IBS. Based on the pooled estimate, coffee drinkers (any intake) may have a decreased risk of developing IBS (pooled OR 0.84, 95% CI: 0.80 to 0.88) compared to controls. The same was observed when comparing coffee drinkers and abstainers (pooled OR: 0.83, 95% CI: 0.79 to 0.87), although the protective effect seems to be driven by a single large cohort study [32].

To the best of our knowledge, this is the first meta-analysis that has examined the association between coffee consumption and the risk of developing IBS. One recurrent theme in ongoing IBS research and discussions is the role of diet. In spite of our meticulous literature search and screening as well as the large number of patients in our pooled analysis, the effect estimates had admittedly low precision. The primary studies were of poor quality and substantial variations existed in the definitions of the exposure variable (coffee intake) and the comparator group used, making it difficult to draw concrete inferences and comment on dose–response or threshold dose.

At a mechanistic level, coffee is a beverage that contains over a hundred compounds, and its contents can vary substantially according to the source, roasting approach, grind and preparation [34]. This heterogeneity in the makeup of what is regarded as coffee was not adequately defined in most of the primary studies reviewed herein. It could account for the lack of an association with IBS in some studies due to compositional variability. For example, Wu et al. [32], who analysed data from a large prospective cohort (the UK Biobank), found a significant dose–response relationship, and the beneficial effect was particularly evident in individuals who consumed instant or ground coffee. Spanning a median follow-up period of 12 years, a notably reduced risk of IBS was correlated with increased coffee consumption. Specifically, individuals consuming 0.5–1, 2–3, and ≥4 cups of coffee daily experienced a 7%, 9%, and 19% reduced risk of developing IBS, respectively. Those who preferred instant or ground coffee had an even greater decreased risk of IBS of 20% [32].

Similarly, studies in liver disease suggest that the preparation method of coffee may influence the effects on disease occurrence, and the effects of coffee may also vary according to the stage of the disease. In morbidly obese women, the consumption of regular coffee is protective against liver fibrosis, but espresso is not [35]. Coffee may decrease the risk of liver fibrosis in fatty liver but is not associated with the incidence of fatty liver [36]. Admittedly, an individual’s coffee or caffeine intake is challenging to quantify and measure consistently across studies. Details like the coffee type, brewing method, and serving size can vary substantially and introduce heterogeneity. This is relevant as the method used to brew coffee plays a role in determining the biochemical makeup and antioxidant content of the resulting beverage; several factors can influence the final composition of the brewed coffee, including the size of the coffee grind, the duration of extraction, the pressure applied, the kind of filter used, the temperature of the water, and more [37]. A study that looked at eight coffee extraction methods (including Espresso, Moka, French Press, Cold Brew, V60, and Aeropress) from both chemical and physical perspectives, all using the same raw material, found that the classic Espresso had the highest concentrations of caffeine and chlorogenic acids, with its extraction efficiency being nearly twice as high as the other methods [37].

In our meta-analysis, the pooled analysis inevitably loses some granularity by simply comparing any coffee drinkers to study controls or non-drinkers because the dose, preparation method, and constituents may influence the biological effects of coffee. There is, unfortunately, a lack of a standardised approach among the primary studies to quantify the actual coffee intake. Future studies on IBS should strive to provide a more granular characterisation of the coffee intake to further elucidate the nature of any links with IBS. The broader health ramifications remain under study.

The potential protective and beneficial effects of coffee drinking on IBS align with some previous studies. According to Wu et al. [32], other constituents present in coffee, including polyphenols, diterpenes, trigonelline, and melanoidins, have been suggested to possibly protect against the development of irritable bowel syndrome (IBS) through their antioxidant and anti-inflammatory properties. Various studies have also demonstrated that substances in coffee can maintain the integrity of the intestinal barrier [38], yield positive effects on the gut microbiota [18,19], and aid in the maintenance of intestinal permeability and mobility, whilst attenuating visceral hypersensitivity and local inflammation. These hypothesised mechanisms could potentially attenuate the pathogenic processes associated with IBS, which include alterations in gastrointestinal motility, visceral hypersensitivity, gut–brain axis dysfunction, low-grade (chronic) intestinal inflammation and impaired epithelial barrier integrity [39,40], thereby reducing one’s risk of developing the condition. There is a biological plausibility of a putative relationship between coffee intake and IBS.

Diterpenes, such as cafestol and kahweol, found in prepared coffee beverages may also have potential helpful effects on gastrointestinal health. Cafestol can suppress bile acid synthesis by downregulating cholesterol 7α-hydroxylase and sterol 27-hydroxylase [41]. Bile acids have been shown to influence the gut bacterial population [16], and high bile acid profiles have been associated with abdominal symptoms, mucosal inflammation, and diarrhoea in a subgroup of individuals with IBS [17].

On the other hand, there are some primary studies that contradict this and report a significant association between coffee intake and an increased risk of developing IBS [29,30,31]. It is important to note, however, that all of these studies were cross-sectional, may contain several biases, and cannot establish causal links between coffee consumption and the risk of IBS. Additionally, two of these studies [30,31] relied on self-reported questionnaires (without further confirmation by a clinician) to assess both exposure (coffee intake) and outcome (IBS diagnosis), which could inadvertently lead to misclassification issues and be susceptible to confounding factors, such as other (undiagnosed) organic colonic disorders. Nevertheless, there is growing interest in the relationship between gut microbiota and various diseases, including IBS [15]. Coffee has been shown to influence the composition of gut microbiota, but the implications of this for IBS risk or symptomatology remain to be fully understood. Studies have documented the effects of diet on gut microbiota and microbial metabolites, although no uniform characteristics of IBS-related gut microbiota have been identified to date [15]. Specific to coffee, previous studies have suggested that coffee intake may be related to the risk of developing IBS due to several proposed mechanisms. Koochakpoor and colleagues [30] discussed the potential link between caffeine intake and the development of IBS. They highlighted that consuming caffeine might disturb the neural control of the gastrointestinal system. This disturbance could be traced back to disruptions in the HPA axis [21,42], which regulate various body functions [43], including stress response. Caffeine ingestion has also been associated with elevated levels of stress hormones like cortisol, epinephrine, and norepinephrine [21,42]. These elevated hormone levels might contribute to the onset and progression of IBS. Furthermore, coffee stimulates gastric acid secretion, which may irritate the intestine, damaging the mucosa and compromising the integrity of the intestinal epithelial tissue [20]. Evidently, the exact mechanism linking coffee to the development of IBS is an area that still requires further research.

There were also several limitations in this review. Firstly, the majority of the included studies are cross-sectional in design, with only one longitudinal cohort study and one case–control study. Hence, the possibility of reverse causality and ascertainment bias from prevalent IBS cases in individual studies must be considered. Secondly, most of the primary studies also had a small sample size without adjustment for potential confounders. The internal validity of these studies is called into question due to inadequate power and residual confounding. The small pool of studies also precluded meta-regression and further subgroup analysis, both in the variety of coffee ingested and the specific IBS subtypes. Moreover, the studies did not collect data pertaining to the specific occurrence of each IBS subtype, even though each IBS subtype has been associated with a distinct bacterial signature [44]. Thirdly, there were wide variations in the reported levels of coffee intake, making it a challenge to establish a standardised approach for categorising the different intake levels. As discussed previously, we resorted to creating a simple binary distinction between coffee drinkers and non-coffee drinkers using available data, which may have obscured a dose–response relationship or a floor effect. Last but not least, most of the studies under consideration originated from Asia, which may limit the external validity of the findings due to the regional and ethnicity-specific variations in one’s gut microbiome. Despite the possibility of a common biological mechanism for coffee on IBS development, the gut microbiome can be influenced by a multitude of factors, including diet, environment, and genetics, which can all be deeply rooted in one’s geographical location and ethnic background [45,46]. There could also be significant inter-individual variations, which translate into inter-individual differences in intestinal metabolism and health effects in response to food [47]. This underscores the importance of personalised dietary recommendations, especially for those diagnosed with or susceptible to IBS [7]. Future studies in this area should try to include diverse ethnic populations for greater external validity.

## 5. Conclusions

This systematic review and meta-analysis found that coffee drinkers may have a reduced likelihood of developing IBS compared to non-drinkers. Still, the conclusion must be interpreted in light of several shortcomings. There is a general paucity of research in this area, and the available studies suffer from several notable methodological issues. The studies also do not consistently demonstrate a beneficial effect of coffee on gut health despite the various bioactive compounds present in coffee. Future research in this area should (1) prioritise high-quality prospective cohort studies with well-documented coffee consumption (and exposure) and track the development of incident IBS in previously healthy individuals over time, and (2) investigate biological mechanisms.

## Figures and Tables

**Figure 1 nutrients-15-04745-f001:**
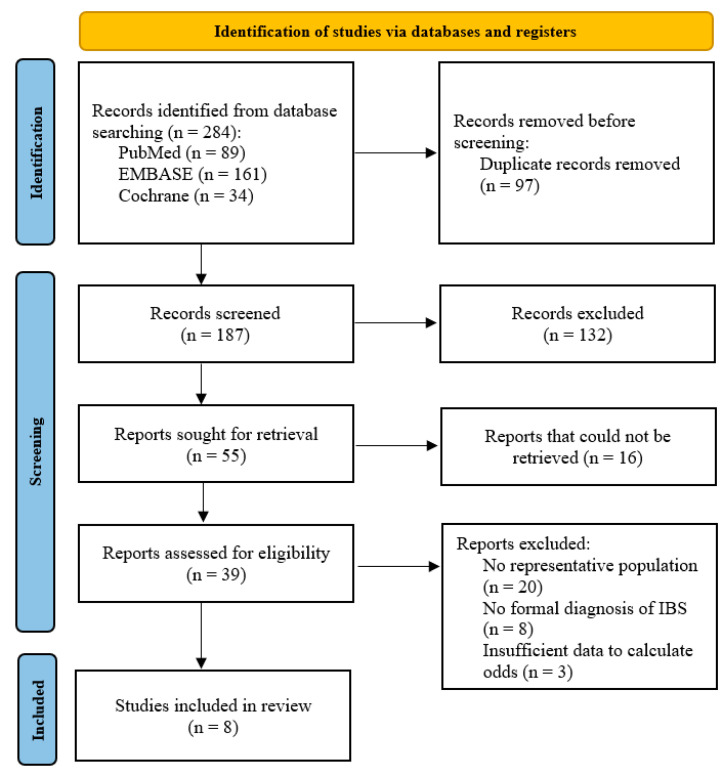
PRISMA flowchart showing the study selection process.

**Figure 2 nutrients-15-04745-f002:**
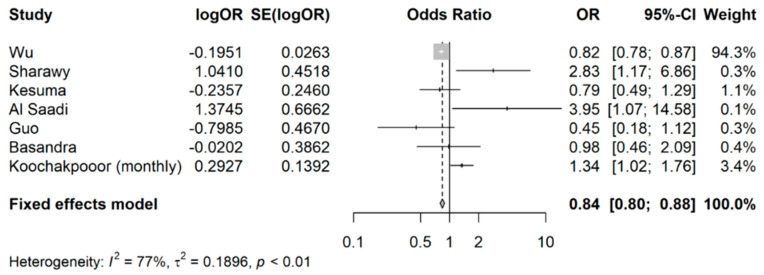
Meta-analysis of OR showing the association between coffee intake (any) and occurrence of IBS.

**Figure 3 nutrients-15-04745-f003:**
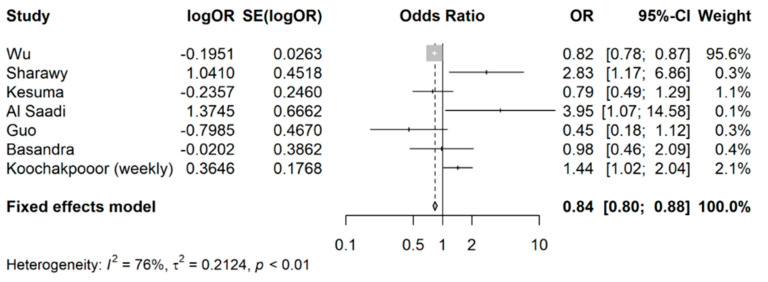
Sensitivity analysis for meta-analysis of odds ratios of association between coffee intake (any) and occurrence of IBS.

**Figure 4 nutrients-15-04745-f004:**
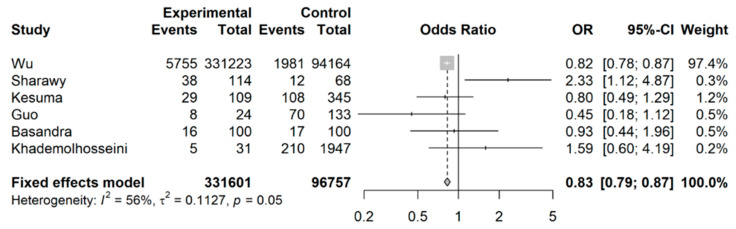
Meta-analysis of binary outcomes (number of IBS cases among coffee drinkers versus non-coffee drinkers).

**Figure 5 nutrients-15-04745-f005:**
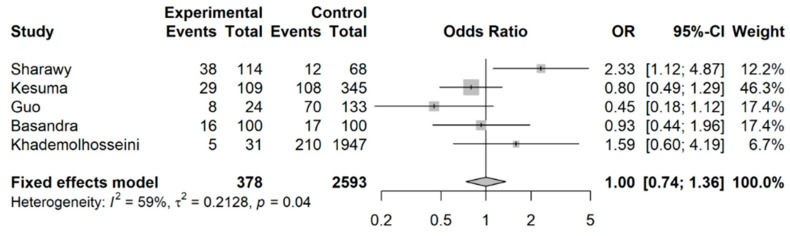
Meta-analysis of binary outcomes (number of IBS cases among coffee drinkers versus non-coffee drinkers), excluding Wu et al. (sensitivity analysis).

**Table 1 nutrients-15-04745-t001:** Characteristics of the studies reviewed, arranged chronologically by year of publication.

Authors, Year	Region	Total Number of Subjects	Total Number of Cases	Study Population		Measure of Coffee Intake	Definition of Coffee Consumption (Lowest vs. Highest Category)	IBS Assessment Criteria
				Sexes Involved in Study	Age of Participants at Baseline (Years)			
Khademolhosseini et al., 2011 [14]	Asia	1978	191	M + F	39	Baseline coffee intake, questionnaire	Consumer vs. abstainer	ROME II
Basandra et al., 2014 [15]	Asia	200	33	M + F	20.4	Baseline coffee intake, questionnaire	0 cups/day vs. >2 cups/day	ROME III
Guo et al., 2015 [16]	Asia	157	78	M + F	46.8/43.3	Baseline coffee intake, questionnaire	Consumer vs. abstainer	ROME III
Al Saadi et al., 2016 [28]	Asia	302	50	M + F	21.6	Baseline coffee intake, questionnaire	1 cup/day vs. >3 cups/day	ROME III
Kesuma et al., 2021 [29]	Asia	454	137	M + F	15.8	Baseline coffee intake, questionnaire	Consumer vs. abstainer	ROME III
Koochakpoor et al., 2021 [30]	Asia	3362	NR	M + F	36.1	Baseline coffee intake, questionnaire	Non-drinker vs. >3 cups/day	ROME III
El Sharawy et al., 2022 [31]	Africa	182	50	M + F	23.9	Baseline caffeine intake, questionnaire	Consumer vs. abstainer	ROME III
Wu et al., 2023 [32]	UK	425,387	7736	M + F	56.22	Baseline coffee intake, questionnaire	0.5–1 cups/day vs. 2–3 cups/day vs. ≥4 cups/day	ICD-10

NR = not reported.

## Data Availability

The datasets analysed or generated during the study are available from the corresponding author upon reasonable request.

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
