# Peer review of "Examining the Association between Coffee Intake and the Risk of Developing Irritable Bowel Syndrome: A Systematic Review and Meta-Analysis"

_nutrients, 2023, doi:10.3390/nu15224745_

Round 1

Reviewer 1 Report

Comments and Suggestions for Authors

Although the meta-analysis itself was conducted methodologically correctly, it seems to me that the discussion and conclusions are insufficient due to the lower quality and unevenness of the studies included in this research. 

The authors themselves emphasize several shortcomings of this study, so attention should be paid to that as well.

Author Response

Thank you for the comments. We have paid attention to the discussion and conclusion sections, mindful of the shortcomings of the primary studies reviewed herein.

Reviewer 2 Report

Comments and Suggestions for Authors

A brief summary

The article is clear, informative and very well systematized. The topic is interesting and could contribute in a scientific sense.

General comments:

Introduction section is very well written. There are some facts in the Discussion section that could be moved in the Introduction for better understanding of the topic. Materials and Methods are also very well and thoroughly described. References should be checked according to the Instructions for Authors.

Specific comments:

Line 40-41            Please rewrite the sentence due to better understanding

Line 47          It is suggested to use more appropriate expression than “healthcare consumption”

Line 55-57            Please state a reference for this fact

Line 64-66            Please state a reference for those impacts

Line 208-211     This was already mentioned in the Materials and Methods section

Line 217-220        These facts belong to the Introduction section

Line 233-248       This paragraph could be shortened or partially transferred to the Introduction section

Line 276-278       It is suggested to perhaps state why their studies showed differently

Line 348                Please address to the “Instructions for Authors” when listing the References       

Comments on the Quality of English Language

The English language should be checked by a professional.

Author Response

Thank you for your comments. We have made some parts from the Discussion section to the Introduction, as suggested, for better understanding of the topic.

  1. For clarity, the sentence (Line 40-41) has now been rephrased to read, "The detrimental impact on quality of life due to Irritable Bowel Syndrome (IBS) is comparable to that caused by organic illnesses, including inflammatory bowel disease [2]. Furthermore, IBS is often associated with diminished work productivity, reduced participation in social activities, and increased instances of absenteeism [3,4]."
  2. We have changed "healthcare consumption" to "healthcare utilisation".
  3. Thank you for the comment. We have added a reference to support the statement at Line 55-57 (citation: Campaniello D, Corbo MR, Sinigaglia M, Speranza B, Racioppo A, Altieri C, Bevilacqua A. How Diet and Physical Activity Modulate Gut Microbiota: Evidence, and Perspectives. Nutrients. 2022;14(12):2456. doi: 10.3390/nu14122456).
  4. Thank you for the comment. We have added a reference for the impact of coffee on the human gut (citation: Nehlig A. Effects of Coffee on the Gastro-Intestinal Tract: A Narrative Review and Literature Update. Nutrients. 2022;14(2):399. doi: 10.3390/nu14020399).
  5. Line 208-211 has now been removed due to redundancy.
  6. Line 217-220 has now been moved to the introduction section.
  7. Line 233-248 has been rewritten in a more concise and succinct manner.
  8. To elaborate on studies that showed contrary findings, we now explained that, "On the other hand, there are some primary studies that contradict this and report a significant association between coffee intake and an increased risk of developing IBS [30,31,32]. It is important to note, however, that all of these studies were cross-sectional in nature, may contain several biases, and cannot establish causal links between coffee consumption and the risk of IBS. Additionally, two of these studies [31,32] relied on self-reported questionnaires (without further confirmation by a clinician) to assess both exposure (coffee intake) and outcome (IBS diagnosis), which could inadvertently lead to misclassification issues and be susceptible to confounding factors, such as other (undiagnosed) organic colonic disorders."
  9. We have referred to the "Instructions to Authors" when presenting our references.

Reviewer 3 Report

Comments and Suggestions for Authors

This is the first meta-analysis that has examined the association between coffee intake and the risk of developing IBS. One recurrent theme in ongoing IBS research and discussions is the role of diet. In the first instance it was proven trated that e, coffee intake was defined rather inconsistently across the primary studies, with some adopting a binary approach (coffee consumer or abstainer) as opposed to 3-level or finer categorisations. In more detail it was found that  individuals consuming 0.5–1, 2–3, and ≥4 cups of coffee daily experienced a 7%, 9%, and 19% reduced risk of developing IBS, respec-tively. Those with a preference for instant or ground coffee had an even greater decreased risk of IBS of 20%. This systematic review and meta-analysis found that coffee drinkers may have a re-duced likelihood of developing IBS compared to non-drinkers. There is a general paucity of research in this area and the available studies suffer from several notable methodolog-ical issues. The studies also do not consistently demonstrate a beneficial effect of coffee on gut health despite the various bioactive compounds present in coffee.

Author Response

Thank you for the comments!

Round 2

Reviewer 1 Report

Comments and Suggestions for Authors

No more comments.